

# Age-related alterations in inhibitory control investigated using the minimally delayed oculomotor response task

Paul C. Knox and Nikitha Pasunuru

Eye and Vision Science, Institute of Ageing and Chronic Disease, University of Liverpool, Liverpool, UK

## ABSTRACT

Healthy, older adults are widely reported to experience cognitive decline, including impairments in inhibitory control. However, this general proposition has recently come under scrutiny because ageing effects are highly variable between individuals, are task dependent, and are sometimes not distinguished from general age-related slowing. We recently developed the minimally delayed oculomotor response (MDOR) task in which participants are presented with a simple visual target step, and instructed to saccade not to the target when it appears (a prosaccade response), but when it disappears (i.e. on target offset). Varying the target display duration (TDD) prevents offset timing being predictable from the time of target onset, and saccades prior to the offset are counted as errors. A comparison of MDOR task performance in a group of 22 older adults (mean age 62 years, range 50–72 years) with that in a group of 39 younger adults (22 years, range 19–27 years) demonstrated that MDOR latency was significantly increased in the older group by 34–68 ms depending on TDD. However, when MDOR latencies were corrected by subtracting the latency observed in a standard prosaccade task, the latency difference between groups was abolished. There was a larger latency modulation with TDD in the older group which was observed even when their generally longer latencies were taken into account. Error rates were significantly increased in the older group. An analysis of the timing distribution of errors demonstrated that most errors were failures to inhibit responses to target onsets. When error distributions were used to isolate clear inhibition failures from other types of error, the older group still exhibited significantly higher error rates as well as a higher residual error rate. Although MDOR latency in older participants may largely reflect a general slowing in the oculomotor system with age, both the latency modulation and error rate results are consistent with an age-related inhibitory control deficit. How this relates to performance on other inhibitory control tasks remains to be investigated.

## INTRODUCTION

The observation that as we age the performance of many tasks becomes more difficult, leads easily to the narrative of inevitable age-related decline. It is unsurprising that this includes the cognitive domain (see *Salthouse, 2012* for a general review). Executive

Corresponding author
Paul C. Knox, pcknox@liv.ac.uk

functions are a set of related abilities by means of which thought and action are controlled. While there is debate about how they are best classified, one approach views them as being separable into three broad domains of task or set shifting, working memory functions and inhibition, based on performance across a range of tasks (*Miyake & Friedman, 2012*; *Miyake et al., 2000*). In this and other accounts, inhibition or inhibitory control is a key domain (*Verbruggen et al., 2014*). The cognitive decline associated with ageing, has been specifically linked to an age-related deficit in inhibition (*Hasher & Zacks, 1988*). However, the hypothesis of a general inhibitory control deficit in older age, particularly once age-related changes in processing speed have been accounted for, is also matter of ongoing debate (*Rey-Mermet & Gade, 2018*; *Rey-Mermet, Gade & Oberauer, 2018*; *Verhaeghen, 2011*, *2014*).

Inhibition itself is not a unitary concept and, as with executive function more generally, a number of taxonomies have been proposed to describe it (*Rey-Mermet, Gade & Oberauer, 2018*). Behavioural inhibition, often investigated with tasks where a motor response has to be prevented or stopped, is a prominent feature of most of these taxonomies. Some tasks used to investigate behavioural inhibition rely on manual responses (e.g. manual stop signal reaction time and go/no go tasks; *Donders, 1969*; *Logan & Cowan, 1984*; *Verbruggen & Logan, 2008*), while others involve eye movements (e.g. saccade countermanding and no go tasks; *Crawford et al., 2005*; *Hanes & Carpenter, 1999*). Note that while these all involve aspects of motor behaviour, a central claim is that this general approach provides a means of exploring cognitive processing related to the control of thought as well as action (*Castiglione et al., 2019*; *Logan & Cowan, 1984*).

A particular eye movement task, the antisaccade (AS) task (*Hallett, 1978*; *Hutton & Ettinger, 2006*; *Munoz & Everling, 2004*), has been widely used in studies of executive function (*Mirsky et al., 2011*; *Miyake & Friedman, 2012*; *Miyake et al., 2000*), healthy ageing (*Fernandez-Ruiz et al., 2018*; *Munoz et al., 1998*; *Olincy et al., 1997*; *Pa et al., 2014*) and to investigate the effect of degenerative neurological conditions associated with ageing (*Crawford et al., 2005*; *Kaufman et al., 2012*). In many studies, the explicit motivation for using the AS task is to investigate inhibitory control (*Alichniewicz et al., 2013*; *Baird-Gunning & Lueck, 2018*; *Crawford et al., 2002*, *2005*; *Diarra et al., 2019*; *Lennertz et al., 2012*). In the AS task, a target appears to the left or right of fixation, but participants are instructed to look to the mirror image position of that target position (i.e. to direct their saccade in the opposite direction but the same distance from fixation). The directional error rate (the proportion of trials in which participants look at the target rather than to the instructed position) is taken to provide a measure of inhibition, with high error rates implying poor inhibitory control.

However, the view that increased AS directional error rate necessarily indicates an inhibitory control deficit is problematic. While successful AS performance requires the inhibition of the normal, reflexive, saccade response to a suddenly appearing target, it also requires the computation and execution of a voluntary saccade to an instructed position. This entails a competition for behavioural expression between two distinct behaviours (the reflexive prosaccade and the voluntary AS), often modelled as a race between two signals rising to a triggering threshold (*Munoz & Everling, 2004*;

*Noorani & Carpenter, 2013*). As whichever process reaches the trigger threshold first 'wins' the race, any difficulty generating the voluntary saccade response will bias the competition in favour of the reflexive prosaccade (error) response, leading to a higher error rate. However, in this case, the increased error rate does not indicate a problem with inhibitory processing (*Reuter, Rakusan & Kathmanna, 2005*). Allied to this, there is the well-recognised issue of task contamination. All tasks engage multiple processes, and this is true of the AS task which requires both attentional (*Gaspelin & Luck, 2018*) and working memory resources (*Crawford et al., 2011*; *Eenshuistra, Ridderinkhof & Van der Molen, 2004*), two other key components of executive function (*Magnusdottir et al., 2019*).

In order to address some of these issues we developed the minimally delayed oculomotor response (MDOR) task (*Knox, Heming De-Allie & Wolohan, 2018*; *Wolohan & Knox, 2014*). The stimulus used is a target step with randomised direction and timing; target display duration (TDD) is also varied. Participants are instructed to execute a saccade to a target appearing on either left or right of fixation, not when it appears (a reflexive prosaccade response), but when the target is extinguished. Performance is measured using the error rate (the proportion of trials in which participants execute a target directed saccade prior to target offset) and the latency of correct responses. We demonstrated that saccade latency in the MDOR task was much longer than is consistent with simple prosaccade responses to target onsets, and that it is modulated by TDD. Latency with a short TDD of 200 ms was of the order of 400 ms. We interpreted this as indicating that saccades in this condition were executed against a high level of inhibition, leading to the considerable increase in latency compared to a standard prosaccade response in which latency would be expected to be approximately 200 ms. With a longer TDD of 1,000 ms, latency was approximately 300 ms, still considerably longer than might be expected. We hypothesised that is was lower because when the offset does not occur quickly, the level of inhibition present when the offset does occur has reduced. These latency effects were of a magnitude that were inconsistent with other known effects such as preview effects, and the lower salience of target offsets compared to onsets.

Minimally delayed oculomotor response error rates are similar to error rates in the AS task, and also modulated by TDD. Analysis of the timing of error responses demonstrated that most errors were consistent with unsuccessfully inhibited responses to target onsets. We have shown previously that AS and MDOR error rates do not correlate in healthy participants (*Wolohan & Knox, 2014*). Manipulating fixation conditions (using gap and overlap task versions) did not significantly alter error rates and latencies in the MDOR task (*Knox, Heming De-Allie & Wolohan, 2018*), although it does in the AS task. The alterations observed in the AS error rate are due to shifting the relative performance of the two competing tasks in the AS context as fixation conditions are changed (*Munoz & Everling, 2004*). In the MDOR task there are no competing processes of this type, and we argued that this explained the lack of gap and overlap effects. While the MDOR task, like every task, is not free of task contamination, the working memory component is greatly reduced compared to the AS task. Although there is an instruction to be remembered (i.e. saccade to the offset not the onset of the target), there is no need to

remember a target position (the target is displayed until shortly before the saccade is initiated). And there is no need to disengage attention from the target position, perform a vector inversion, and shift it to a new location, as would be required for a correct AS.

All of the experiments we have conducted with the MDOR task so far have been on younger participants. Across multiple groups of younger participants, we have observed performance to be similar and consistent, noticeably so for error rate (*Knox, Heming De-Allie & Wolohan, 2018*). To explore the MDOR task further, and given the prominence of saccade tasks in both the ageing and inhibitory control literatures, our aim in the current study was to investigate MDOR performance in older healthy participants, and to compare it to that in younger participants.

## METHODS

### Ethics and participants

A consecutive, convenience sample of healthy, older, adult participants, with normal or corrected to normal vision was recruited from the local community, under ethical approval from the University of Liverpool Research Ethics committee (Reference No. 2933). We did not conduct a formal power calculation. All participants provided informed, written consent after the experiment was explained to them and they had an opportunity to ask questions. Participants were offered £10 to compensate them for their time and for the expense of travelling in to the University for testing. Comparison data were available from younger participants from previous experiments. We identified participants aged between 18 years and 28 years tested previously on the same MDOR task, using the same equipment and procedures.

### Apparatus and stimuli

We used the same apparatus and stimuli as in earlier experiments (*Knox, Heming De-Allie & Wolohan, 2018*; *Wolohan & Knox, 2014*). Briefly, stimuli were presented on a 21′ monitor (1,024 × 768 spatial resolution, 100 Hz temporal resolution) driven by a VSG2/5 card (Cambridge Research Systems, Rochester, UK), positioned on the fronto-parallel plane 57 cm from the participant's eye. Horizontal eye position of the left eye was recorded using a Skalar Iris IR Eye Tracker, with the eye tracker output digitised with 16 bit precision using a CED Power 1401 (Cambridge Electronic Design, Cambridge, UK) interface. Oculomotor data was stored for off-line, trial-by-trial analysis using custom software.

We used the synchronous version of the MDOR task (*Knox, Heming De-Allie & Wolohan, 2018*; Fig. 1) in which a central fixation target ($0.2°$ black square; $11 \text{ cd/m}^2$) was presented on a light background ($56 \text{ cd/m}^2$; contrast 79%) for a randomised period of 0.5–1.5 s. Immediately when it was extinguished, the saccade target (a $0.2°$ black square) appeared $5°$ to the left or to the right of fixation (randomised and with equal frequency) and was displayed for either 200 ms or 1,000 ms (randomised from trial to trial). Participants were instructed to maintain fixation in the centre of the display until the saccade target disappeared, when they were to execute a saccade to the target's location (i.e. saccade on target offset), pause, and return their gaze to the centre of the

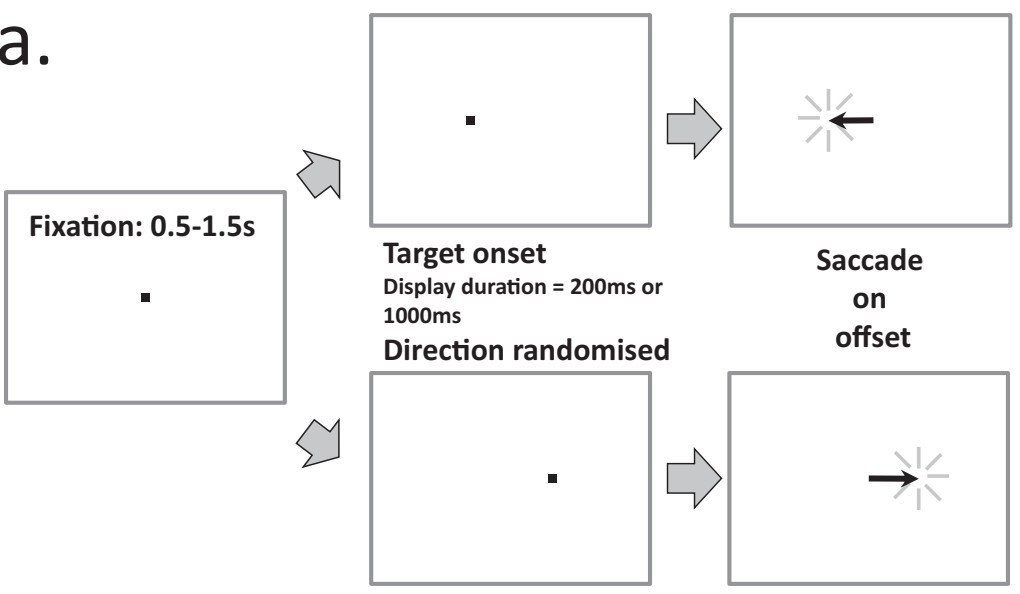

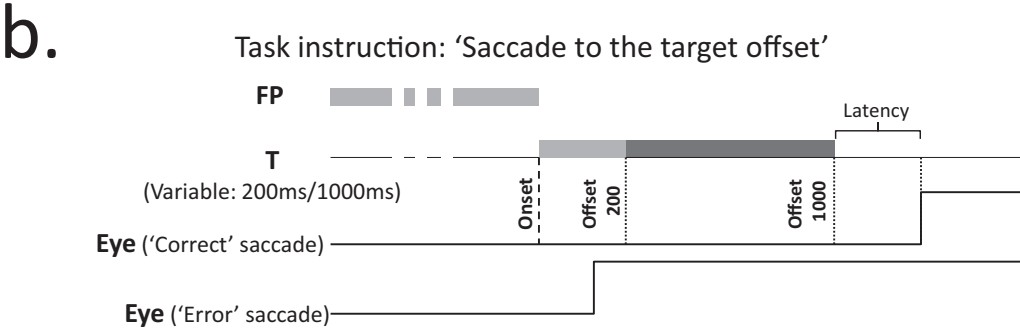

**Figure 1 Description of the MDOR task.** (A) Trial structure. After a randomised fixation period of 0.5–1.5 s, the target appeared 5 degrees to either the left or right. Participants were instructed to look to the target position when it disappeared (i.e. saccade on offset). (B) Trial and response timings. Targets were displayed for either 200 ms or 1,000 ms. The example correct saccade, executed in response to a target displayed for 1,000 ms, occurs after the offset. The example error saccade occurs soon after the onset, well before either of the possible offsets.

display in preparation for the next trial. They were explicitly instructed not to saccade to the onset of the target. Before testing commenced, the trial sequence was demonstrated to participants; this was repeated until they were happy that they understood what was required.

## Procedures

Participants were carefully positioned by adjusting table height, a chin rest and cheek pads. They completed two runs of 120 MDOR trials usually with a break in between them. The quality of performance was carefully monitored to ensure that it was maintained, with verbal feedback given as necessary. A 32 trial calibration procedure was performed after each run. In calibration trials, after a randomised fixation period (0.5–1.5 s), the fixation target was extinguished and a saccade target was presented to the left or right at an

eccentricity of 5° or 10° (randomised and with equal frequency) for 1 s. Participants were instructed to fixate the central point and saccade to the target as soon as it appeared, fixating it until it was extinguished, at which point they could blink and return to the centre, ready for the next trial. These procedures were identical to those used in earlier experiments. Older participants also completed the Addenbrookes Cognitive Examination (ACE) III questionnaire (*Hsieh et al., 2013*).

## Analysis

Oculomotor data were analysed using an interactive programme which displayed the eye position data and the time at which the 'go' signal (target offset in MDOR trials) occurred. The calibration data were used to transform the data from arbitrary system units into units of degrees of eye rotation, allowing us to measure saccade amplitude. Trials with blinks or unstable fixation prior to target appearance were removed from the analysis. For each valid trial (i.e. trials free of blinks and poor fixation), the latency and amplitude of the primary target-directed saccade were measured.

   Data were collated in MS Excel. Error responses were first identified, removed and collated separately from correct trials, and the error rate calculated. Any target directed saccade with an amplitude greater than 1° that occurred from 80 ms after target onset to 80 ms after target offset was counted as an error. Saccades occurring <80 ms after target onset were classified as anticipatory saccades, in common with many other studies (*Amatya, Gong & Knox, 2011*; *Wolohan & Knox, 2014*), and not included in further analysis. Any target directed saccade occurring from 80 ms to 1,000 ms after target offset was counted as a correct response. Note that in comparison data for younger participants taken from a previously published study (*Knox, Heming De-Allie & Wolohan, 2018*), this period was 80 ms to 600 ms after target offset. In the current data, this additional 400 ms period accounted on average for 2.5% of correct responses.

   For each participant, median saccade latency was calculated along with the error rate. Across participants, both latency and error rate were summarised using the mean. Error and latency data for the two TDDs (200 ms and 1,000 ms) were kept separate. We used the median saccade latency from calibration data, collapsed across direction and eccentricity, to provide an estimate of reflexive prosaccade latency, rather than use control MDOR runs. This reduced the testing time for older participants, and similar (calibration) data were available for younger participants tested previously. Statistical analysis (details in "Results") was conducted with SPSS v22.

## RESULTS

Twenty-two older participants (13 male) were recruited and tested in the present study. The mean (±SD) age of this group was 62 ± 7 years (range: 50–72 years). Eight participants were aged 50–59 years, eight 60–69 years and six 70 years and over. The mean ACE III total score for these older participants was 96 ± 3 (range 91–100). Figure 2 illustrates both individual and group latency and error rate data for these participants. Across the older group the mean number of trials available for analysis per participant was 200. The mean proportion of valid trials contributing to the analysis was 83% (i.e. 17% of

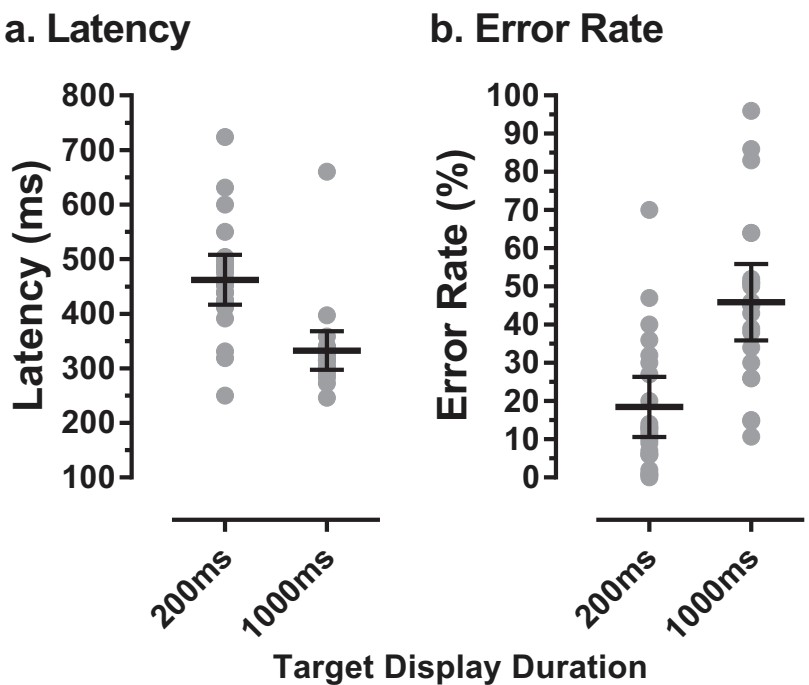

**Figure 2 MDOR latency and error rate in older participants.** (A) Latency, (B) error rate, observed in 22 older participants. Grey data points show individual results; horizontal bars show group mean ±95% CI.                                                

trials were lost to blinks, poor fixation, or had no responses). As in previous experiments with the MDOR task, latency was longer than would be expected for prosaccade responses, and modulated by the TDD. In tasks with a 200 ms TDD, mean group latency was 462 ± 103 ms compared to latency in the calibration task of 236 ± 57 ms. For the 1,000 ms TDD, mean latency was 332 ± 80 ms. Similarly, for error rate we observed the same general pattern as in previous experiments; for a TDD of 200 ms the mean error rate was 18 ± 18% and for a TDD of 1,000 ms it was 46 ± 23%.

MDOR task performance in the older group was compared to that in younger participants tested previously (*Knox, Heming De-Allie & Wolohan, 2018*; *Wolohan & Knox, 2014*). We used data from a group of 39 younger participants, with a group mean age of 22 ± 2 years (range 19–27 years; 15 male). The mean number of trials available per participant was 187 (slightly lower than for the older group), and the mean proportion of trials analysed per participant was 78% (i.e. a rejection rate of 22%). The difference in the per-participant proportion of trials available for analysis between the two groups was not statistically significant ($t = 2.0$; df = 54; $p = 0.22$). When compared to this younger group, there was an apparent ageing effect on raw latency (Fig. 3A). For the 200 ms condition the latencies were 462 ± 103 ms and 394 ± 61 ms in old and young groups respectively (a difference of 68 ms) while for the 1,000 ms condition they were 332 ± 80 ms and 298 ± 46 ms (a difference of 34 ms). When tested with a repeated measures ANOVA with group (old, young) as a between and TDD (200 ms, 1,000 ms) as a within-subjects factor, both factors returned statistically significant results

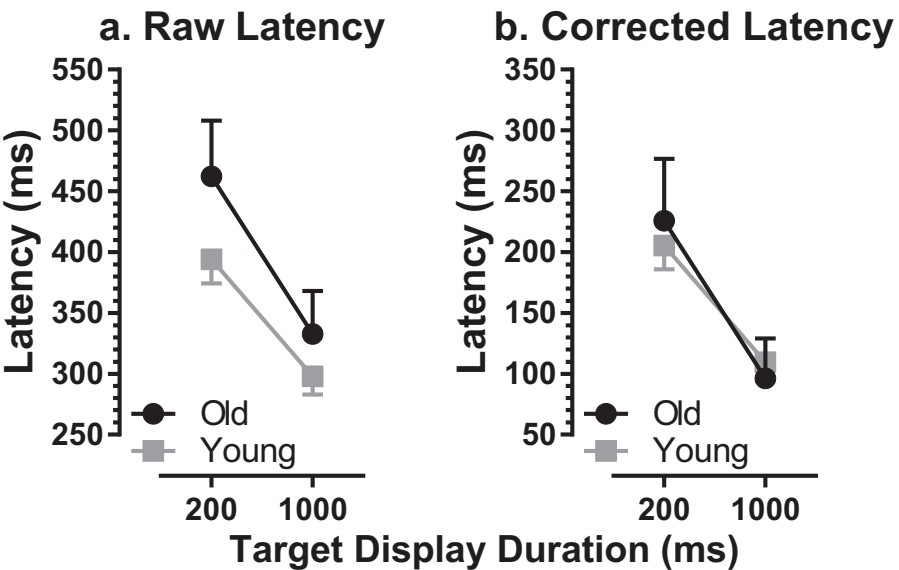

**Figure 3 Comparison of MDOR latency in old and young participants.** Comparison of latency between old and young groups. Mean (±95% CI's) is shown. (A) Raw latency. (B) Corrected latency. Latency in the calibration task is subtracted from each participant's MDOR latency.

(group: $F_{1, 59} = 14.1$, $p < 0.001$, $\eta_p^2 = 0.19$; TDD: $F_{1, 59} = 78.0$, $p < 0.001$, $\eta_p^2 = 0.57$) with no statistically significant interaction ($F_{1, 59} = 1.73$, $p = 0.19$).

We also calculated effect size using Cohen's $d$ (sometimes called $d_s$ as the sample rather than population means and standard deviations are used, with Bessel's correction; *Lakens, 2013*). For each participant we calculated the mean of the latencies for the two TDDs, and then calculated the group mean and standard deviation (old: 397 ± 55 ms; young: 346 ± 49 ms). Cohen's $d$ was 0.99. Given the relatively small sample size, and that this can artificially inflate $d$, we also calculated the corrected effect size (Hedge's $g_s$). However, the correction factor was only 0.000088 (i.e. $d$ was left essentially unchanged).

There was a significant difference in latency between old (236 ± 57 ms) and young (189 ± 41 ms) groups for reflexive prosaccade latency, which was available from the calibration task ($t = 3.4$; df = 33; $p = 0.002$; $d = 0.92$). For each participant we modified the MDOR latency by subtracting their calibration latency, to take account of general age-related changes in the saccade circuitry. This abolished the latency difference between groups (Fig. 3B).

Given that the group data suggested the modulation of latency by TDD might differ between the two groups (although the interaction term in the ANOVA was not statistically significant), we calculated the difference in latency between the 200 ms and 1,000 ms conditions for each participant and then summarised for the groups. The mean difference was 164 ± 81 ms in the older group and 96 ± 47 ms in the younger group, statistically significantly different ($t = 3.4$; df = 26; $p = 0.002$; $d = 1.12$). To account for the generally longer latencies in the older participants, we also calculated this difference as a percentage of the average latency of the two TDDs. This yielded a difference of 40 ± 15% for the older

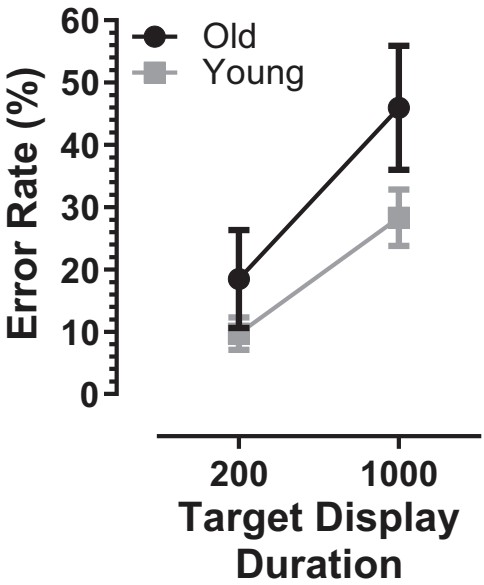

**Figure 4 Comparison of error rate in old and young participants.** Mean (±95% CI) MDOR error rate (%) in old and young participant groups.

and 28 ± 12% for the younger group; these differences remained statistically significant ($t = 3.2$; df = 33; $p = 0.003$; $d = 0.92$). The data of two older participants with very high error rates and therefore very few correct trials (who did not exhibit the usual pattern of higher latency in the 200 ms condition) were removed from this analysis.

MDOR error rates were higher in the older group compared to younger participants (200 ms condition: 18 ± 18% vs. 10 ± 8%; 1,000 ms condition: 46 ± 23% vs. 28 ± 14%; Fig. 4). Using a repeated measures ANOVA similar to that described above (group as a between and TDD as a within subjects factor) both group ($F_{1, 59} = 12.6$, $p = 0.001$, $\eta_p^2 = 0.18$) and TDD ($F_{1, 59} = 200.7$, $p < 0.001$, $\eta_p^2 = 0.77$) returned statistically significant results. In contrast to the results with raw latency, the interaction between group and TDD was also statistically significant ($F_{1, 59} = 7.4$, $p = 0.009$, $\eta_p^2 = 0.11$). Investigating the group difference for the two TDD's further using post-hoc $t$-tests and the sequential Holm–Sidak procedure to control for Type I errors, we found that the difference in error rate for the 200 ms condition did not reach statistical significance ($t = 2.19$, df = 26; corrected $p = 0.073$) while the error rate for the 1,000 ms did ($t = 3.31$, df = 30, corrected $p = 0.002$). Cohen's $d$ for the error rate was 1.01.

The distribution of both error and correct response timings was explored by plotting the pooled distributions for both old and young groups of participants for both TDDs (Fig. 5). For both groups, there were clear peaks after target *onset* in a latency range consistent with them being failed attempts to inhibit the response to target onset (time = 0 ms; indicated by the vertical dotted lines in Fig. 5). These peaks occurred in the same latency bins (bin width = 50 ms) for both old and young participants (the −50 ms bin for the 200 ms condition, Figs. 5A and 5C; the −850 ms bin for the 1,000 ms condition, Figs. 5B and 5D). For both conditions, these bins are the fourth bin after target appearance,

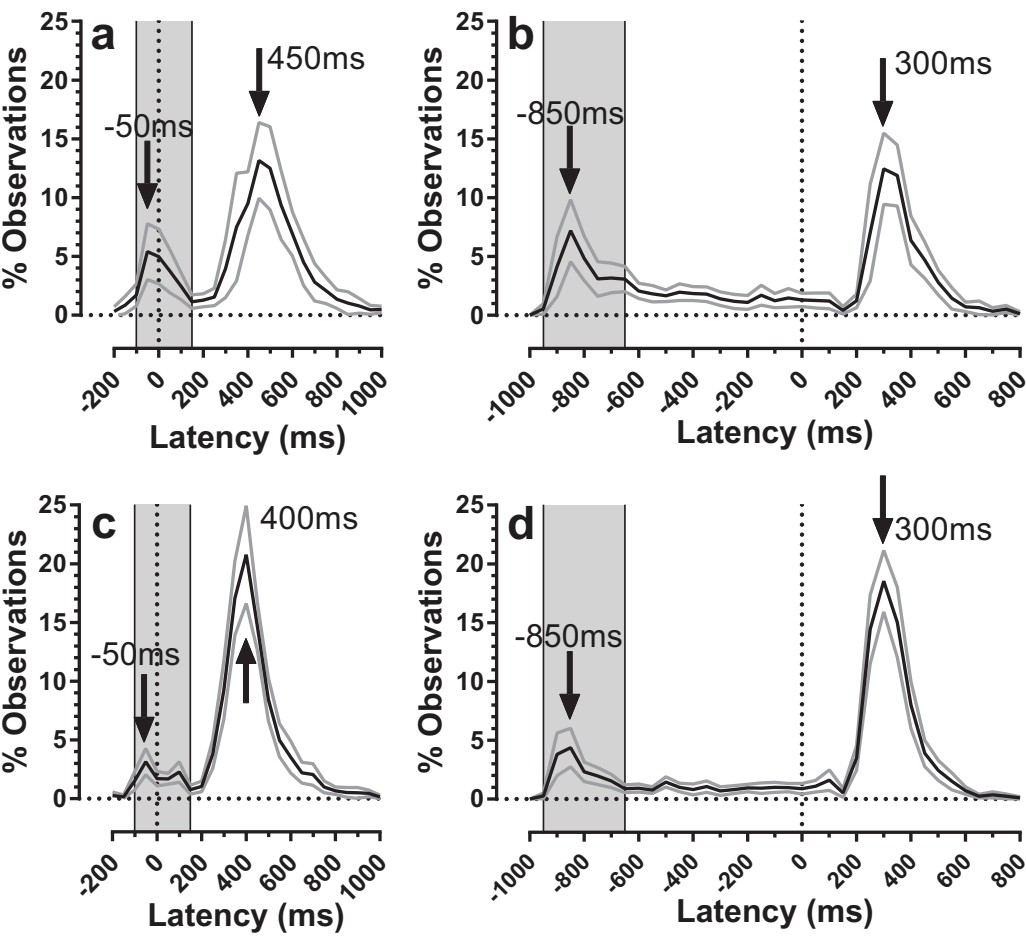

**Figure 5 Average distributions of response timings for old and young participant groups.** Average distributions are shown for old (A and B) and young (C and D) participant groups. Bin width = 50 ms. For each participant, the % frequency distribution for responses was calculated. The mean (±95% CI) was then calculated for each bin across participants in each group. The black central line is plotted through the each mean bin value, while grey lines show ±95% CI. Target onset is at −200 for 200 ms TDD (A and C) and −1,000 ms for 1,000 ms TDD (B and D); target offset (the go signal) is at 0 ms (vertical dotted line). Arrows mark peak bins in each distribution for both errors and correct responses. Grey regions mark the bin ranges over which error rate was recalculated.

comprising responses with latencies in the rage 150–199 ms. We defined latency ranges which captured these early peaks (for the 200 ms condition: −100 ms to 150 ms; for 1,000 ms: −950 ms to −650 ms; Fig. 5) and recalculated error rates to include only responses in these ranges. This had relatively little effect on error rates for the 200 ms condition (old: 20 ± 18%; young: 11 ± 8%), and a more marked effect on error rates in the 1,000 ms condition (old: 26 ± 20%; young: 15 ± 12%). We retested these data with the same design of ANOVA as above; group ($F_{1, 59} = 8.1$, $p = 0.006$, $\eta_p^2 = 0.121$) and TDD ($F_{1, 59} = 12.3$, $p = 0.001$, $\eta_p^2 = 0.121$) returned statistically significant results, while the interaction was no longer significant ($F_{1, 59} = 0.7$, $p = 0.41$). We computed the effect size ('$d$') for TDD's of 200 ms and 1,000 ms separately and these were 0.67 and 0.72 respectively. For the 1,000 ms condition we also examined the residual error rate defined as the

proportion of responses occurring from −600 ms to 0 ms. This was again higher for the older group (21 ± 10% vs. 12 ± 7%; $t$ = 3.6, df = 35; $p$ = 0.001; $d$ = 1.02).

## DISCUSSION

Studies of oculomotor control have been widely used over a long period to investigate the effects of ageing (*Abel, Troost & Dell'Osso, 1983*; *Morrow & Sharpe, 1993*; *Olincy et al., 1997*; *Simons & Buttner, 1985*; *Spooner, Sakala & Baloh, 1980*). In general, as might be expected, oculomotor performance declines with age. The effect of ageing on AS performance has also been widely studied, with there being general agreement that AS latency and error rate both increase with age (*Butler, Zacks & Henderson, 1999*; *Olincy et al., 1997*). AS task performance is often taken to provide an index of inhibitory control, and therefore its decline with age is assumed to support the hypothesis of a general age-related deterioration in inhibitory function (*Hasher & Zacks, 1988*). However, while clearly involving inhibitory control, the AS task involves other key aspects of executive function, and AS directional error rate is influenced by multiple factors (*Bowling, Hindman & Donnelly, 2012*; *Lee et al., 2010*; *Magnusdottir et al., 2019*). Our motivation in developing the MDOR task was to target oculomotor inhibitory control more precisely, and in the present study to investigate the effects of normal ageing on MDOR performance.

We have discussed previously the differences between the MDOR task and other types of saccade tasks, such as the memory guided saccade task (MGS) which it resembles (*Knox, Heming De-Allie & Wolohan, 2018*). In the MDOR task there is no extended memory delay (typically a delay of several seconds in the MGS task), the target is present throughout the fixation period and the go signal is provided by a transient at the target position, not fixation. The MDOR task also resembles other delayed saccade tasks, which lack the working memory load of MGS tasks (*Hutton et al., 2002*; *Reuter et al., 2007*). However, there are important differences in detail in between studies relating to the timing of the removal of the central fixation target, and the nature and timing of the go signal (e.g. an auditory tone was used in *Reuter et al. (2007)*, not a target offset). Such methodological differences should be borne in mind when comparing studies as they are likely to affect both error rate and latency.

We have also discussed previously other effects which might contribute to the pattern of results observed in the MDOR task (*Knox, Heming De-Allie & Wolohan, 2018*; *Wolohan & Knox, 2013*). The magnitude of the latency increases we have observed are much larger than can be accounted for by preview effects, or the lower salience of target offsets compared to target onsets. It is in tasks in which oculomotor responses have to be inhibited that latency increases of this magnitude are observed (*Machado & Rafal, 2000*).

More generally two broad types or classes of inhibitory control have been defined. Proactive inhibitory control involves top-down, preparatory inhibition before responses are initiated, whereas reactive inhibitory control involves the inhibition of an initiated or prepotent action, often in response to a cue or signal (*Aron, 2011*; *Verbruggen et al., 2014*). The classic AS task (in which blocks of only AS are run) contains both proactive and reactive aspects, a complication noted by *Aron (2011)* in his classification of tasks

(e.g. see his Table 1). The MDOR task is perhaps more proactive in nature in that in every trial in a run or block the emphasis is on the participant not executing a saccade to the target when it appears. Nevertheless, there is still a reactive element in that, as with the AS task, the appearance of the target (the external 'cue') triggers saccade programming in each trial, and it is this that must be inhibited.

We recruited a group of healthy (by self-report) older participants and screened them with the ACE III questionnaire. All scored above the cut-off for suspicion of neurological disease (usually quoted as a total score of 88) and the group mean was very similar to that reported in *Hsieh et al. (2013)* for a similarly aged control group. MDOR performance in our older participants was qualitatively similar to that observed previously (*Knox, Heming De-Allie & Wolohan, 2018*; *Wolohan & Knox, 2014*). Latency in correct trials was much longer than is consistent with reflexive prosaccades, and was modulated by TDD. We did not run a control MDOR task in the current experiment as we have done previously, in part to reduce the time commitment demanded of our participants and to avoid participant fatigue. However, the calibration task that was run, which is identical to that which we have used in previous studies, is a prosaccade task, with randomised target direction and randomised fixation duration. It therefore provides a means of obtaining an estimate for prosaccade latency in our participants. Comparing MDOR latency with that observed in the calibration task, latency was increased by 226 ms in the 200 ms TDD condition over calibration task latency, and by 96 ms in the 1,000 ms condition; the magnitude of these increases is similar to those reported when the comparison is with an MDOR control task (see *Knox, Heming De-Allie & Wolohan, 2018*; Table 1).

Latency was increased in the older compared to younger participants, by an average of 68 ms in the 200 ms condition and 34 ms in the 1,000 ms condition. These are of an order that might be expected given ageing effects on latency in other saccade tasks. What is of more interest is the extent to which these differences are specific to the MDOR task (and therefore indicative of changes in inhibitory control in ageing), or simply a reflection of a general slowing in the oculomotor system (*Verhaeghen, 2011*). For prosaccade tasks, latency increases with age, although the reported increases are variable, dependant on factors such as the exact age of participant groups, and the type of task used. The difference of 47 ms observed between our young and old groups on the calibration prosaccade task is similar to that reported elsewhere for similarly aged groups (*Eenshuistra, Ridderinkhof & Van der Molen, 2004*; *Klein et al., 2000*).

When we corrected MDOR latency for both groups by subtracting prosaccade latencies from MDOR latencies, the latency difference between groups was abolished, suggesting that most if not all of the latency increase in the MDOR task could be due to general slowing in the saccade system in older participants. A similar result has been reported for ASs (*Butler, Zacks & Henderson, 1999*). Given the differences between the calibration and MDOR tasks, this result should be treated with caution. The calibration task was not simply a repetition of the MDOR task without inhibition, as is the MDOR control task.

Increased latency in the MDOR task is indicative of the requirement to inhibit the reflexive response to target onset (discussed in detail in *Knox, Heming De-Allie & Wolohan, 2018*). The modulation in latency with TDD is consistent with higher levels of inhibition if the target offset (the go signal) occurs soon after target onset (as in the 200 ms condition), with the level of inhibition declining the longer the target is displayed. In pilot studies we have observed a monotonic relationship between latency and TDD tested over a wide range (*Knox & Abd Razak, 2010*). The larger latency modulation in the older group, which was still present once the generally longer latency in the older group was accounted for, suggests either a higher level of inhibition initially, a lower level of inhibition eventually, or a combination of both, compared to the younger group. This could be compatible with the older participants exerting a great inhibitory effort as a strategy for improving performance overall, with the implication that this cannot be maintained as effectively as in the younger participants, leading to a larger difference in latency between the two TDD conditions used here.

Higher MDOR error rates in the older group relative to the younger group parallels a similar finding in the AS task (*Abel & Douglas, 2007*; *Klein et al., 2000*; *Peltsch et al., 2011*). We have shown previously that although both the AS and MDOR tasks involve the inhibition of reflexive prosaccades, the error rates in the two tasks do not correlate when compared in the same participants (*Wolohan & Knox, 2014*). On this basis, closely comparing error rates between the two task types is probably of little value. However, while they may not correlate it is worth noting that in general error rates in these two tasks are similar, and that interparticipant variability in older participants tends also to be increased for both tasks.

While none of our participants reached a 100% error rate in the current study, one older participant reached 96% in the 1,000 ms condition. As with the AS task, precise error rates are influenced by a variety of factors including the details of task design, numbers of trials run, block structure and so on, many of which are yet to be investigated for the MDOR task. But bearing in mind that one use of the MDOR task might be in patient populations, in which higher rates might be anticipated compared to the healthy participants studied here, using a version with generally lower error rates, and reduced variability, would be an advantage. Both error rate and latency in the AS task are modified by changing fixation conditions. In many clinical AS studies this means that a gap AS task is used to avoid floor effects (i.e. 0% error rates) in control groups. However, this option is not available with the MDOR task (*Knox, Heming De-Allie & Wolohan, 2018*). But investigating other stimulus and task features (e.g. the provision of more spatial information by using placeholders, the use of three TDDs as opposed to two) might generally improve performance without fundamentally altering the ability of the task to provide insights into inhibitory control.

The error rates that we calculated captured all errors (defined as responses occurring between >80 ms post target onset and <80 ms after target offset) as a proportion of the total number of valid responses. However, the average distributions (Fig. 5) suggest that only some of these errors are uninhibited responses to the target onset. These are the responses that make up the early peaks, around the −50 ms/−850 ms bins in Fig. 5.

Recalculating the error rate to capture only these early responses perhaps provides a purer measurement of behavioural inhibition in the MDOR task. On this basis, error rate remained significantly higher in older participants, implying weaker inhibitory control. This measure might have the advantage of avoiding ceiling effects in participants with known problems with inhibition, particularly for longer TDDs. However, this does raise the issue of how to classify the other errors which occurred at a low rate throughout the fixation period in the 1,000 ms condition. This 'residual error rate' was also significantly higher in the older participants. Conceptually it seems plausible that the residual error rate represents a different aspect of inhibitory function. While it is possible that these errors are also uninhibited responses to target onsets, those occurring at long latencies might reflect the ability of participants to maintain fixation (i.e. maintain inhibition) in circumstances where a particular action is anticipated. In this context it is worth noting that neither in the current or previous experiments have we found evidence of a significant peak towards the end on the fixation period in the 1,000 ms condition that might represent target offset anticipations. This suggests that with only two TDDs (and randomisation of direction) there is sufficient uncertainty in the task to prevent participants successfully anticipating the target offset.

Given the clear difference between groups in error rates, it appears that even if older participants adopt a different strategy by, for example exerting a maximal inhibitory effort at target onset (which they find difficult to maintain at long TDDs), reflected in increased latency, they still cannot inhibit their oculomotor responses as successfully as younger participants. How this relates to other aspects of inhibition remains to be seen. While a wide range of tasks have been used to investigate inhibition, correlations between them are absent or low (*Friedman & Miyake, 2004*; *Rey-Mermet, Gade & Oberauer, 2018*). This is often taken to be due to the general problem of task impurity (which we have also argued is a problem with the AS task). The MDOR task is clearly not immune to exactly the same problem. There is a working memory load in the MDOR task (e.g. there is an instruction to be remembered). However, it is reduced compared to the AS task. Nor would it be true to say that there is no attentional load in the MDOR task. But as the target position for the response is the same as the position of the target onset, there is no requirement to disengage, shift and re-engage spatial attention at a non-target location, as there is in the AS task. So the observed effects are more closely tied to the ability of participants to inhibit their responses.

## CONCLUSION

The MDOR task provides a useful means of studying oculomotor inhibition. Latency increases in the MDOR task with age appear to primarily reflect general age-related slowing of the saccade system. However, the higher MDOR error rates observed in the older group, particularly for those errors consistent with uninhibited responses to target onsets, confirms an inhibitory control deficit in normally ageing participants.

## ACKNOWLEDGEMENTS

We are grateful to all the participants who took part in these experiments.

### Funding

This work was supported by the MRes Clinical Sciences programme in the University of Liverpool. There was no additional external funding received for this study. The funders had no role in study design, data collection and analysis, decision to publish, or preparation of the manuscript.

### Grant Disclosures

The following grant information was disclosed by the authors:
MRes Clinical Sciences programme in the University of Liverpool.

### Competing Interests

The authors declare that they have no competing interests.

### Author Contributions

- Paul C. Knox conceived and designed the experiments, performed the experiments, analysed the data, prepared figures and/or tables, authored or reviewed drafts of the paper, and approved the final draft.
- Nikitha Pasunuru performed the experiments, analysed the data, authored or reviewed drafts of the paper, and approved the final draft.

### Human Ethics

The following information was supplied relating to ethical approvals (i.e. approving body and any reference numbers):

Ethical approval was granted by the University of Liverpool Research Ethics Committee (Reference No. 2933).

### Data Availability

The data is available at Figshare: Knox, Paul (2019): MDOR Ageing Data-Latency IQR's added. figshare. DOI 10.6084/m9.figshare.11310527.v1.

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
