# Peer review of "Age-related alterations in inhibitory control investigated using the minimally delayed oculomotor response task"

_PeerJ, doi:10.7717/peerj.8401_

## Round 0.1 · original submission · Major Revisions

Both reviewers have reported significant problems with the data analyisis and how results are presented. Please check these aspects before the next submission.

Reviewer 1 ·

Basic reporting

Knox and Panusuru investigated the oculomotor inhibitory control by employing a task that required the subjects to make a saccade when a visually presented target disappear (after a delay of 200 or 1000ms since its appearance). This task involves inhibitory control because it requires subjects to refrain from making a saccade as soon as the target appears in their visual field. Making a saccade towards a target as soon as it appears is typical behavior in humans. Overall the paper is well written.
However, while the task has the potential to investigate oculomotor control, I have some comments and suggestions that I hope will help to improve the manuscript.

Introduction:
Lines 46-47. Please rephrase “In the AS task…target position”. In the phrase, “mirror image” makes difficult to understand the task. Better to write "the opposite direction" or something similar.

Experimental design

Most of the concerns are, at the moment, of experimental nature, related to the methods and analysis. The analysis seems too complicated. I suggest the authors employ a simpler design to analyze the data. In the current version of the paper, too much emphasis is done to data and results (f.e. stratified analysis in results) that do not reach significance. I have some concerns also about the sample size for old subjects (n=20), but by adding the effect sizes, the possible limitation can be at least be evident to the readers.
Furthermore, it is not really clear to me after reading the paper if the differences between young and old are showing a difference in inhibitory control or a “general slowing of the saccadic system”, as stated by the authors, line 287 and in the conclusion. It seems that the most important results are hidden in the text to give an immediate take-home message.
Methods.
Ethics and participants.
Line 93. Comparison data were available from younger participants from previous experiments.
Is it necessary to use some statistical correction? Did the authors use it?

Apparatus and Stimuli
Line 111. “They were explicitly instructed not to saccade to the onset of the target.”
Which is the complete instruction that they provided to the subjects?

Procedures
General: To help the self-sufficiency of the paper, I think it is better to add the description of the experimental procedure employed with the “young group” too.
The authors compared 39 young subjects to 22 old subjects (although I recommend to reduce the sample to 20, because two subjects performed very poorly, see below for line 195). Why did they do not collect data from a higher number of old participants? Please motivate this choose (it can also be a practical reason).
Line 104. Which is the amount of the step in the fixation randomized period (0.5-1.5s)?
Line 114. Please add the duration of the break between sessions.
Lines 114-116. “The quality of the performance was carefully monitored…[…]…as necessary.” What does “as necessary means”? Which were the criteria to provide feedback?
Lines 116-119. As I understood the experimental sequence was: Run-Calibration-Run-Calibration.
Did the authors use some form of training for the subjects before the experiment started? Did they discard the first trials of the first Run sequence? (I guess that subjects will be learning a bit the task in this phase).
Are the initial calibration trials different from the last calibration trials? I am asking this because I would expect some “Switching” like effect, related to the different rules applied for oculomotor control in the two contexts. This effect could be different between young and old subjects.
Please, add a figure that illustrates the task.

Analysis
Line 32 How are the saccades between 0 and 80 ms after target onset classified?
Lines 134-135 “Similarly…[..]..as a correct response”. The authors employed an interval of 80-1000ms to collect correct saccades. However, they compare these data with those of young people in which the interval was 80-600ms (Knox et al. EBR 2018). I am wondering if such a difference could have affected the results. Did the subjects receive some feedback while performing the task? How much time were they allowed during the trial to make a saccade? Related to this topic, I did not understand (but I am sure I am missing something) how it is possible to have few subjects with a median latency very close to 600ms in Figure 2 (Knox et al. EBR 2018, right upper panel (MDOR sync)).

Line 135 “For each participant …mean error rate“. Why did the authors employ the median for the latency and the mean for the error rate?
Line 138 “ We used median saccade latency collapsed ..”. Is there a difference between latencies for the two targets eccentricities? If this is the case, I think the 5° should be a better option.

Validity of the findings

Results
The main goal of the paper is to investigate the age-related alteration in inhibitory control using the minimally-delayed oculomotor response. To this aim, it is necessary to compare the performances of the old-subjects with the performances of the young subject.
To make it simpler I would suggest employing an ANOVA design with groups (old, young) as between factor and tasks (calibration, MODR (with two levels(TDD): 200 and 1000ms))as a within-subjects factor.
This can be applied to both errors and latencies.
Alternatively, it is possible to run a Group x TDD design twice: one with the real values, the other correcting for the calibration values.
The same design should be used for the last analysis performed (Ref. Figure 6).
Line 157 The stratified analysis does not report any real effect. As such Figure 2 and 3 can be deleted. I suggest the authors run an exploration analysis by using a regression model.
Line 195: “The data of two older participants…”. These subjects must be excluded from the beginning. The sample of the old subjects should be reduced to 20.
Line 206 The variability of the older group can be just related to the higher variability of the age. I suggest the authors calculate the intra-subjects variability (see f.e. Pani et al. 2013 RDD) to make the comparison. Typically the variance is linearly related to RT, but it does not change always consistently.
I also suggest the authors, if this is possible, to calculate some after-effects. For example what happens to the latency of the saccade when an error is committed in the previous trial? I do not know if this can be done, because it is not clear to me if subjects had some feedback for every early saccade or the classification was done only after data collection.
Effect sizes should be included.
Data files:
In the. xls file provided, would be useful to have a measure of variance (standard deviation or inter-quartile range together with the median latencies.

Additional comments

I also think that the discussion must be re-written. Once the new analyses are done, and the results are briefly reported, I suggest to make some comparison not only with the anti-saccade task but also with the stop-signal task, considering its diffusion (Verbruggen et al. 2019 Elife). Furthermore, while the topic of the paper is inhibition, I think that other processes must be accounted for. For example, shorter latencies for longer TDD reminds me of the delay preparatory effect. As such the effect observed by the authors can be due to a higher level of movement preparation. This is not necessarily against the decline in inhibition, but maybe it is related to the nature of the delay task and to the interaction between movement generation and inhibition often observed ( Duque, Greenhouse et al. 2017 TICS). Furthermore another topic that can be discussed is the proactive inhibitory control (Elchlepp H, Lavric A, et al. 2016 Cogn, Psych.; Pouget, Logan et al. 2011 Jour of Neurosci; Lo CC, Boucher et al. 2009, Jour of Neurosci; Wardak C, Ramanoël S,et al 2012 EJN). Furthermore, related to the fixation activity see for example Stevenson SA, Elsley JK, Corneil BD (2009) J Neurophysiol.
I am not asking the authors to cite all these works, but I think they can be of help for the discussion.


AS stated above, I think the discussion must be rewritten once the analyses are done.
Here I add just some comment on sentences that grabbed my attention:

Lines- 242-244: Our motivation…target inhibitory control more precisely …[..].
The inhibitory process always involves different stages (see Verbruggen et al. 2014, Perspect Psychol Sci.), so caution must be taken when talking about inhibition as a unitary concept or a simple process(see also Salum GA, Sergeant J, et al (2014) Psychol Med 44; Montanari Giamundo et al. EBR 2017; Indrajeet and Ray 2019 EJN). I think this topic must be explicitly discussed.

Lines 283-289. In the discussion, the authors refer to the stratification analysis to support their argument. But the results lack statistic significance. This part must be removed.
Line 291- 292 (discussed in detail in Knox et al. 2018). I think the paper must include all the arguments necessary. Please, give at least a summary of the detailed discussion to make the paper “self-sufficient”.
Lines 291-303: please consider also the preparatory effect in accounting for reduced latency with increasing delay.

Reviewer 2 ·

Basic reporting

In their paper, Knox and Pasunuru used the MDOR task to test for inhibitory processes in older adult population. The MDOR task has the advantage, contrary to the antisaccade task (AS), of not involving other higher cognitive processes, hence, giving a more ‘pure’ measure of inhibitory control. They observed that not only saccade latency was increased in the MDOR task compared to reflexive prosaccade task, but also that error rates were higher in the older population compared to younger adults. Majority of these errors in the older population was due to a failure to inhibit reflexive saccade. The authors conclude that not only MDOR task showed a classical slowing of the oculomotor system with aging, it also showed deficit in inhibitory control.
The article is clearly written and I am enthusiastic about the MDOR task developed by Knox and colleagues and the idea that this task could be used to assess for age-related deficit of inhibitory control. Although the introduction could benefit from some improvement, the major weakness of this paper is the results section (see comments in section 3).

Major comments:
1) If found that problematic very interesting, but I also found that the authors go a bit too fast on the link between aging and inhibitory control. The introduction could benefit from more emphasis on why inhibitory processes are important to study, particularly in aging, which would lead to how do we assess for this and why MDOR task should be preferred to AS task.In this regard, I would suggest developing a bit more around the aging thematic. Some information could be added to the second paragraph where the authors only state “a range of task has been used to investigate the effect of aging on behaviourally inhibitory control”. But the notion of inhibitory control is not defined, nor is the notion of executive functions. I would suggest to briefly give a definition of executive functions (EFs), then maybe briefly state the basic processes included in the EFs. Then focus on inhibition, how do we define it, how we generally assess for it (e.g., go no-go tasks (e.g., Potter & Grealy, 2008), stop-signal tasks (e.g., Rush, Barch, & Braver, 2006), negative priming tasks (e.g., Hasher, Stoltzfus, Zacks & Rypma, 1991). And then, go onto AS task (paragraph from L69).

2) Line 90/91 the authors state that the problem related to AS task is the fact that it involves attentional and working memory resources, which are believed to be important mechanisms for executive functions. But a reader not familiar with EFs, can wonder why this is an important issue, especially for aging. If the authors define what are the EFs and what processes they include from the very beginning, that it would be easier to follow the logic of why MDOR could be a better alternative to AS.

3) In the next paragraph, the authors sum up their previous findings using the MDOR task with younger population. They summarize their observation, and report that “Saccade latency in the MDOR task is much longer than is consistent with simple prosaccade responses to target onsets, and modulated by TDD”. Although the sentence itself needs rewriting, I would also like the authors to develop a little bit more here, give the reader some explanation why is that.

4) Line 105, the authors underline the observation that neither a gap nor an overlap affect error rates or latencies in the MDOR task, contrary to AS tasks. Why? The authors give an explanation in their 2018 paper, which could be beneficial here.

Minor comments:
L. 80 - “a manual version of the AS task […] complicating the interpretation of the results”.
I would like the authors to develop a bit more here
L 110 - Could the authors state the age of the participants? Also “multiple groups” – can the authors specify how many.

The results section structure could be improved with adding subsection as for example:
1. Latency & Error rates
2. Latency & error rates by group age
3. Comparison between young and old groups
4. Distribution of error and correct responses timings

Figures are clear although they need some improvements :
Figure 2 - Error bars are missing.

Figure 5 & 6 - Only the upper limit for the aged group and the lower limit of the young group of the CIs are shown

Experimental design

As stated before, I found the research question here very interesting and meaningful.
The method section could benefit from minor improvements.

I would rather have the number of participants and age in the participant sub-section of the methods than at the beginning of the result section.

L 125 “We used the same apparatus[…]” Can the authors give the reference of the publication here?
I would also like to have a figure of the paradigm, I suggest using figure 1 from the 2018 paper.

L 133 “Light background” It can sound a bit meticulous, but how light is light? Can the authors give the value in cd/m²?

L 157 please specify how many trials were rejected for 1)blinks, 2)brake of fixation.

L 158 “valid trials” – is it “correct” trials? I would find informative the average accuracy in both the MDOR and the calibration task (see also comment above), could the information be put either in a table or in a supplementary material?

Validity of the findings

As stated above, my major issue regarding the manuscript is the result section. My first issue is related to the latency results and the comparison between the MDOR task and the control/calibration phase. Not only I feel that the author should report the latency/accuracy of the saccades in the calibration phase in more details, but I also think that the way they compute what they refer to as “corrected latency” is not really correct. Second, regarding the analysis according to the age is also problematic, first given the few numbers of participants per group and also given the large variability in the error rate. Thirdly, the comparison between young and old group should also be taken cautiously, as the number of participants in each group is quite different (n = 39 for the young, n = 22 for the old). I will develop these three points below.

Major comments:

In the first paragraph of the results, the authors briefly report the average saccade latencies (SL) and error rates, and state that on average SLs were longer in the MDOR task than would be expected in the prosaccade task. To prove their point they report the average SLs in the MDOR task separately for 200 ms TDD and 1000 ms TDD, and the average SL in the calibration task. I do not doubt about the fact that the MDOR task leads inevitably to an increase in SL. But there are numerous differences between these two tasks that make them hardly directly comparable.
Contrary to the previous experiment, the authors did not use a control task which they justify by the fact that they did not want the experiment to last too long, especially with older participants. Thus, they used the data from a calibration phase to measure for the prosaccade latencies and calculate the ‘corrected latency’, which is the difference between latency in the MDOR task and the latency in the calibration task. I understand the rationale here but directly subtracting the latencies from the calibration to the latencies from the MDOR poses different problems.

1) The numbers of trials between the two tasks are very different. The authors collected 240 trials for the MDOR task, and only 64 trials for the calibration phase, which is 4 times less. To be able to directly compare the two measures; the number of trials should be about the same between the two tasks.

2) The authors do not report any selection criteria for the calibration phase. If the calibration phase is here to be considered as the baseline for prosaccade latencies, then some selection criteria should be used (amplitude, anticipation, direction…).

3) In the MDOR task, the target was always presented at 5° whereas in the calibration, it could be at 5 or 10°. The author stated that they collapsed all the data between eccentricities and direction to measure for reflexive saccade latency. Before doing that, they should ensure that there was no direction effect and no significant differences in latencies between 5° and 10°. I would also like to see the data for this calibration phase reported in a table. Same comment for the TDD, the TDD of 200 ms and 1000 ms were only present in the MDOR task, and not the calibration.
Given the three problems described above, I do not find the corrected latency measure valid. If the authors want to have a pure measure of how long participants are in the MDOR task compared to prosaccade tasks, then they should do a proper control task with the same eccentricity, TDD and the same number of trials.

4) Looking at Figure 1a and 1 b, it looks like in the 200ms TDD participants are longer but they also show fewer errors, and reversely, in the 1000 ms, participants are faster but show also more errors, which is confirmed by the average values reported in the 1st paragraph of the results section. How about the landing position? Are the participants more accurate in the 200 ms?

5) As one can see on figure 1b, one participant showed more than 90% or error and two other about 80% of error. This high error rate suggests that these participants did not perform the task properly, hence, only very few trials were kept for the latency analysis for these participants. On the other hand, some participants showed close to no error at all, especially in the 200 ms TDD. But in figure 1a we can see that some participants show very long latencies, about 700 ms for one participant in particular. I was wondering whether this participant showing very long latencies in the correct trials was also the one showing lowest error score. Same apply for the high error score: is the one showing more than 90% of errors is the one with the shortest latency in the correct trials.

6) Regarding the data analysis stratified according to age, as the authors concede, the number of participants in each group is very small, between 8 and 6 making results hard to interpret. The observation that latencies of the older group were higher compared to the two other ones is not surprising. However, I would have thought that this group would also exhibit more errors, which is not the case. Regarding the high variability observed on figure 1b, will it be possible that the participants showing between 90 & 80 % or errors were part of group 2? As the errors bars are not reported on the graphs, it is hard to tell.

7) Regarding the comparison between young and old groups, the number of participants in each group is pretty different – 39 participants for the young group and 22 for the older group, which is clearly not ideal. Which makes me wonder how the authors decided on the number of participants for this second experiment? Besides, in the comparison between old and young groups, the corrected latency in the later has been made using a control task.

8) Regarding the distribution of error and correct responses as a function of response timing, the authors first took the distribution of all trials (whether the response is correct or incorrect). We can clearly see bimodal distributions, with one peak appearing early (which corresponds to errors) and a second peak appearing later (which corresponds to correct responses). Correct me if I am wrong here, but the error rate was recalculated– instead of taking 80 ms before/after target, they used 100 / 150 ms for the 200 ms and -950 / -650 for the 1000 ms. They report the average error rate using this new calculation and performed an ANOVA on the remaining data. I found this analysis clever, although I would rather suggest doing the distribution for each individual separately. I would then measure the number of error in the first peak ‘bin’ for each individual and then take the average number of errors. Also, I do not see the separate plots for the error and the correct responses?

I found the discussion clear, the authors nicely point out the limitations of the MDOR task and suggest how to improve it.
As for the introduction, some part of the discussion could benefit from more details. For example, L274 the authors stated “We have discussed previously”, I understand that there is no need to re-discuss the issue here. However, one or two sentences summarizing the discussion could be added. Same regarding the “other delayed saccade tasks” (L276).

Minor comments:
Results
L 166 “none of these differences reached statistical significance” - can the authors please give the reader the F value here?
L 202 “tested previously” – please give the reference here
L208 – with the F values, can the authors add the effect size?
L 215 - The authors looked at the difference between TDD separately for each group. Then they measure the “difference as a percentage of the average of the two TDD”. I don’t get the measure, how is it calculated?
L 232 - The authors observed a significant interaction, can they report any post-hoc analysis?
L 234 – I do not find that the calculation of the CoV really adds anything to the results already reported
L 240 – Can the authors specify the bin used (50 ms?)

Discussion
L 262 “performances decline with age” – specify to which type of performances the authors refer here
L 288 “Latency […] (TDD) – suggest rephrasing
L 298 “These […] saccade tasks” – can the authors give some references here
L 304 – add some references here as well

Additional comments

My suggestion would be to either get rid of the 'corrected saccade' data - just report the average latency and accuracy in the calibration phase in a table and describe the data as part of the first paragraph (same as it is now). In turn, this will restrain the conclusions you can make regarding the increase of SL in the MDOR task, which is quite problematic.
The other suggestion, which would be really great but I understand much more costly, would be to add a second experiment with increasing the number of participants so as to have about 36 participants in total, 12 participants for each group age. This not only will increase the number of participant per group for the stratified analysis but will also reduce the difference with the young group, allowing a better between groups comparison. I would also suggest adding a proper control, all the more if the authors want to assess for an increase in latency in the MDOR compared to prosaccades.

---

## Round 0.2 · Minor Revisions

Additional minor changes are still needed before final acceptance. Please see Reviewers comments.

Reviewer 1 ·

Basic reporting

no comment

Experimental design

no comment

Validity of the findings

Dear Editor,
Dear authors,
here are my comments on this new version.


--Previous revision:

1.6 General: To help the self-sufficiency of the paper, I think it is better to add the description of the experimental procedure employed with the “young group” too.
The authors compared 39 young subjects to 22 old subjects (although I recommend to reduce the sample to 20, because two subjects performed very poorly, see below for line 195). Why did they do not collect data from a higher number of old participants? Please motivate this choose (it can also be a practical reason).

R1.6 The experimental procedures (stimuli, tasks, equipment, instructions) employed for the young participants were identical to those used for the older group. The comment added in response to R1.5 further emphasizes this.


-And the second reviewer wrote:
2.24 Regarding the comparison between young and old groups, the number of participants in each group is pretty different – 39 participants for the young group and 22 for the older group, which is clearly not ideal. Which makes me wonder how the authors decided on the number of participants for this second experiment? Besides, in the comparison between old and young groups, the corrected latency in the later has been made using a control task.
R2.24 The 39 younger participants were those from pervious experiments who were aged 19-27y. There seemed to no good reason to restrict ourselves to the same number as the older group. The N for the older group was a convenience sample as a starting point.



--How did they establish the “convenience”? Did they perform a power analysis? Please state it in the paper.


--Previous revision:

1.14 Line 32 How are the saccades between 0 and 80 ms after target onset classified?

R1.14. These are classified as anticipations, not uninhibited responses to the onset. There is an extensive literature of saccade latency ranges for different type of saccade (eg see Amatya et al. 2011; Knox and Wolohan 2014).

--Please add this to the text.


Previous revision:
1.15 Lines 134-135 “Similarly…[..]..as a correct response”. The authors employed an interval of 80-1000ms to collect correct saccades. However, they compare these data with those of young people in which the interval was 80-600ms (Knox et al. EBR 2018). I am wondering if such a difference could have affected the results.

R1.15 The period from 600ms to 1000ms post-target offset accounts for an average of 2.5% of responses in the older group, and about 1.9% of responses in the younger group (a difference of 0.6%). We do not believe this materially affects the results.

--Please add this to the text.



--Results.

--Line 245 and in other lines…,
please add the df for the t-test


--Line 266 reference must be to Fig. 3b

--Line 268.
“Given that the group data suggested the modulation of latency by TDD might differ between
the two groups (although the interaction term in the ANOVA was not statistically significant),we calculated the difference in latency between the 200ms and 1000ms conditions for each participant and then summarised for the groups. The mean difference was 164±81ms in the older group and 96±47ms in the younger group, statistically significantly different (t=3.4; p=0.002; d=1.12).”


--Honestly, I cannot see how “the group data suggested” considering that the interaction between factors is not significant and that in line 285 they found an interaction effect for the errors but the authors did not analyze it.

--Furthermore I do not understand how it is possible that the effect that they are looking for (the size of the modulation in latency related to the age) it is not detected by the interaction in the ANOVA, but it is detected by looking at the differences.
The part in the discussion (lines 401-408) should be more cautious. I don’t see a strong effect here if it is detected by the t-test on the differences and not by the ANOVA.

--Line 285. Please perform the post-hoc on the interaction.

--I am wondering if the main effect between groups it is driven by the difference in error rates between old and young just for the long TDD.

--Furthermore, please add a reference to figure 4.


--Line 289. I wrote a comment in the previous revision.

--Previous revision:
1.24 Line 206: The variability of the older group can be just related to the higher variability of the age. I suggest the authors calculate the intra-subjects variability (see f.e. Pani et al. 2013 RDD) to make the comparison. Typically the variance is linearly related to RT, but it does not change always consistently.

--And the authors replied:

R1.24 The CoV has been used in previous studies of aging effects on oculomotor control (eg Munoz et al, 1998, Exp Br Res 121:391) and provides useful comparative information. We could not find the suggested reference with the information supplied.


--I agree with the second reviewer that what the authors wrote, line 288
“For both latency and error rate, the results for the older group appeared to be more variable than for the younger participants. This was checked by calculating the coefficient of variation (CoV). For latency, CoV was consistently higher in the older group for both the 200ms condition (22% vs 15%) and the 1000ms condition (24% vs 16%). There was less of a difference for error rate (200ms: 96% vs 82%; 1000ms: both 49%)”, does not add anything to the results, at least in this form.

--Furthermore, they refer to the paper of Munoz et al. 1998, but they present a different analysis. They present the results as percentages (and this is not clear to me).
In the previous revision, I suggested looking at the intra-subject variability (not at the inter-).
It does not matter to me if they use the CoV instead of the SD; indeed Munoz et al. 1998 looked at the intra-subject variability:
“ From the data of each subject, we computed the following values in the eight experimental conditions representing the factorial combination of saccade task (anti vs. pro), fixation condition (gap vs. overlap), and target direction (right vs. left): the mean saccadic reaction time (SRT) for correct trials; the coefficient of variation of SRT for correct trials [(CV=standard deviation/mean)*100]; the percentage of express saccades (latency: 90±140 ms; Fischer et al. 1993); and the percentage of direction errors (saccades away from the target in the pro-saccade task; saccades toward the target in the anti-saccade task).”
Furthermore, in the same paper, pag. 395
“To ensure that the measure of variance was not biased by variations in mean SRTs between age groups, we used the coefficient of variation (CV). The CV was lower in the gap condition (mean CV: 26.1) than in the overlap condition (mean CV: 27.1) [F(1, 167)=5.85, P<0.0001 due in part to a reduced express mode leading to a tighter anti-SRT distribution”.

So I would ask the authors to perform the same analysis as in Munoz et al. 1998. They have to run the same ANOVA performed on latencies and error rates on the CoV.

And again, the inter-subject variability that they report can just be an epiphenomenon related to the ages of the groups. In the old group (as in the xls file added by the authors) the SD is 7.28 years; in the young group, it is 1.89 years.

Previous revision:
1.27 Data files:
In the. xls file provided, would be useful to have a measure of variance (standard deviation or inter-quartile range together with the median latencies.

R1.27 We don’t propose to change the supplied information.

I cannot imagine a valid opposition to this request.
The authors have calculated the CoV; thus add the columns (4!) should be very easy.

Additional comments

no comment

Reviewer 2 ·

Basic reporting

I am really positive regarding the use of eye-tracking in aging studies, and as I previously mentioned I found the MDOR task very interesting. The authors made clear responses regarding my first set of comments and made good points regarding some objections I had previously. Besides, I really appreciate the improvements made in the results section. I still have few comments regarding the methods and results section.

Experimental design

In my first set of comments, I asked the authors to specify the number of rejected trials separately for blinks and break of fixation. I understand that the authors did not record these separately, however, just for replicability purpose, can the authors specify their criteria for "poor fixation" (or "unstable fixation"). My guess would be that saccadic eye movements made during the random fixation period were discarded, if so, what was the saccade amplitude criteria here? Was it 1° as well?

Instead of describing the analysis as a "repeated measure anova" I suggest to rather use the term of "mixed anova" as there is one between and one within-subject factor here.

As previously noted in my first review, regarding the interaction reported on l.285, the authors responded that further analysis "would not add anything". The interaction itself indicates that indeed, old and young participants showed different error rates to the TDDs - and this is showed on Figure 4. However, I still think that further analysis to determine the exact nature of the interaction should be reported here.

I am not sure that only showing one of the two CI error bars is really correct. I understand that bars might be overlapping and that the authors prefer not to display them for clarity. However, CI bars are informative and even though the authors suggest that this information can be seen from figure 6 (which is figure 5 in the revised manuscript), both CI should be shown on the line graphs as well. To get around the clarity issue, the author could use shadows instead to represent +/- 95% CI.

The figures referred to in the result section do not correspond to the figures reported (e.g., l 266 - fig. 4b should be fig.3 b, l281- fig5 should be fig.4).

Validity of the findings

no comment

---

## Round 0.3 · Minor Revisions

I kindly ask the authors to reply to all the (previous and current) comments of the reviewer. In the case of a 'preference not to answer' the decision can be considered acceptable only if sufficiently justified.

Reviewer 1 ·

Basic reporting

none

Experimental design

none

Validity of the findings

none

Additional comments

Two main points (A and B):
A- In the rebuttal letter the authors skipped this point:
--Line 268.
“Given that the group data suggested the modulation of latency by TDD might differ between
the two groups (although the interaction term in the ANOVA was not statistically significant), we calculated the difference in latency between the 200ms and 1000ms conditions for each participant and then summarised for the groups. The mean difference was 164±81ms in the older group and 96±47ms in the younger group, statistically significantly different (t=3.4; p=0.002; d=1.12).

--Honestly, I cannot see how “the group data suggested” considering that the interaction between factors is not significant and that in line 285 they found an interaction effect for the errors but the authors did not analyze it.
--Furthermore I do not understand how it is possible that the effect that they are looking for (the size of the modulation in latency related to the age) is not detected by the interaction in the ANOVA, but it is detected by looking at the differences.

They just replied to this last part:
The part in the discussion (lines 401-408) should be more cautious. I don’t see a strong effect here if it is detected by the t-test on the differences and not by the ANOVA.
I kindly ask them to reply to the first part too.

B- Raw data.
I asked to add values in the raw data (4 columns) that should be easily available (because in the previous version of the paper they estimated the CV), but they wrote that this would be time-consuming. I am sorry but I cannot accept this position, in a period in which data availability is a synonym of transparency and is requested almost everywhere.
If the authors do not make these data available I will not consider the paper ready for publication.

---

## Round 0.4 · accepted · Accept

Thank you for responding to all comments received. The quality of your manuscript has certainly improved and the impact will benefit.

Reviewer 1 ·

Basic reporting

I am satisfied with the answers.
The paper, in my opinion, is now ready for publication.

Experimental design

none

Validity of the findings

none

Additional comments

none